# Energy and Cost Analysis and Optimization of a Geothermal-Based Cogeneration Cycle Using an Ammonia-Water Solution: Thermodynamic and Thermoeconomic Viewpoints

**Nima Javanshir [1], Seyed Mahmoudi, S. M. [1,*], M. Akbari Kordlar [1] and Marc A. Rosen [2]**

[1] Faculty of Mechanical Engineering, University of Tabriz, Tabriz 51666-16471, Iran;
Nimajavanshir@gmail.com (N.J.); Mehri.akbari@tabrizu.ac.ir (M.A.K.)

[2] Faculty of Engineering and Applied Science, University of Ontario Institute of Technology,
Oshawa, ON L1G 0C5, Canada; Marc.Rosen@uoit.ca

* Correspondence: s_mahmoudi@tabrizu.ac.ir

**Abstract:** A cogeneration cycle for electric power and refrigeration, using an ammonia-water solution as a working fluid and the geothermal hot water as a heat source, is proposed and investigated. The system is a combination of a *modified Kalina cycle (KC)* which produces power and an absorption refrigeration cycle (ARC) that generates cooling. Geothermal water is supplied to both the KC boiler and the ARC generator. The system is analyzed from thermodynamic and economic viewpoints, utilizing Engineering Equation Solver (EES) software. In addition, a parametric study is carried out to evaluate the effects of decision parameters on the cycle performance. Furthermore, the system performance is optimized for either maximizing the exergy efficiency (EOD case) or minimizing the total product unit cost (COD case). In the EOD case the exergy efficiency and total product unit cost, respectively, are calculated as 34.7% and 15.8$/GJ. In the COD case the exergy efficiency and total product unit cost are calculated as 29.8% and 15.0$/GJ. In this case, the cooling unit cost, $c_{p,cooling}$, and power unit cost, $c_{p,power}$, are achieved as 3.9 and 11.1$/GJ. These values are 20.4% and 13.2% less than those obtained when the two products are produced separately by the ARC and KC, respectively. The thermoeconomic analysis identifies the more important components, such as the turbine and absorbers, for modification to improve the cost-effectiveness of the system.

**Keywords:** ammonia–water binary working fluid; absorption refrigeration; Kalina; power and cooling cogeneration; thermoeconomic; optimization

## 1. Introduction

In recent decades, rising concerns over fossil fuels shortages and environmental impact have motivated investigators to seek energy conversion methods with high efficiency and low environmental impact. Cogeneration systems have received much attention because of their promising features such as high thermodynamic and environmental performance as well as low production cost [1]. Also, renewable energies including geothermal, solar, wind and biomass are being increasingly exploited because of their sustainability and abundance [2]. Among the types of renewable energy, geothermal waters, due to their large quantities and high stability, are increasingly prevalent and are likely to play an important role in future energy systems [3,4].

Binary mixtures such as ammonia-water solutions are used in many energy conversion systems because of their variable boiling point (approximately −60 °C to −10 °C), allowing the solution to reach an acceptable temperature balance between the solution and components, resulting in lower exergy destructions in components such as heat exchangers, absorbers and boilers [5]. In parallel, many studies have been carried out on cogeneration, tri-generation and multi-generation systems employing binary mixtures such as the ammonia-water solutions [6,7]. To utilize low-temperature heat sources, the organic Rankine cycle (ORC) and the Kalina cycle (KC) have both been demonstrated to be reliable and applied extensively in electricity generation [8–14]. To produce cooling, absorption chillers are widely utilized. They are driven by a heat source, which can be industrial waste heat or solar thermal energy [15].

Much work has been done on cogeneration systems. A distinct cogeneration cycle, capable of generating cooling and electrical power simultaneously, was suggested by Goswami [16]. The proposed cycle operated with an ammonia-water working fluid and one heat source. Said et al. [17] experientially investigated a solar driven cogeneration system using an ammonia-water solution as a working fluid and achieved a COP of about 0.7 and 10 kW refrigeration capacity. Cao et al. [18] comprehensively evaluated a combination of the KC and an ARC by applying a genetic algorithm to achieve the maximum exergy efficiency. Seckin [19] investigated a power/cooling cogeneration cycle which combined KC and ejector refrigeration cycles. In that study, the effects were examined on the produced power and cooling as well as first and second law efficiencies of the cogeneration cycle of varying several parameters, including basic ammonia-water concentration, turbine inlet pressure, condenser temperature and refrigerant pressure in the heat exchangers. Also, the effects of several refrigerants on the performance of the cogeneration cycle are investigated. In order to produce electrical power and cooling simultaneously via heat recovery from the exhaust gases of a gas engine, Chen et al. [20] proposed and examined a cogeneration cycle. In that study, the exergy efficiency and produced net power were found to be 33.7% and 92.9 kW respectively.

Despite the fact that most discussed combined cooling and power (CCP) cycles exhibit satisfactory outcomes, in terms of the energy and exergy efficiencies, these factors do not guarantee whether or not the system is cost-efficient. A different and superior approach known as the exergoeconomic (or thermoeconomic) analysis has been used in many novel studies [21,22]. From this point of view, exergy and cost analyses are combined, reflecting both thermodynamic as well as economic perspectives simultaneously.

Rashidi and Yoo [23] compared the Kalina power/cooling cogeneration cycle (KPCC) and the Kalina $LiBr - H_2O$ absorption chiller (KLACC) cogeneration cycle from thermodynamic and exergoeconomic viewpoints. The exergoeconomic results demonstrated that the unit cost of production for the KPCC is 20.5% less than the corresponding value for the KLACC cogeneration cycle. Akbari and Mahmoudi [24] conducted a thermoeconomic analyses of a combined power and refrigeration cycle comprising organic Rankine and absorption refrigeration cycles. They optimized the cycle and concluded that the production unit cost is 20.4% less in the optimized case than the original case. Ahmadzade et al. [25] conducted thermodynamic and thermoeconomic analyses and optimization of a solar-driven combined Rankine and ejector refrigeration cycles, using an ammonia-water working fluid. The total production unit cost was seen to be 7.8% less for the optimized case. Shokati et al. [26,27] investigated with exergoeconomics absorption power and cooling cogeneration cycles based on the KC and including an ammonia-water double effect KC/ARC and two configurations of the KC and ARC. They concluded that the boiler and low-pressure absorber make the highest contribution to the exergy destruction and capital investment cost rates, implying that these components are the least cost-efficient components in the proposed system.

Thermodynamic and thermoeconomic analyses are conducted in order to determine the energy and exergy efficiencies and unit exergy cost for each stream. Additionally, the effect of varying decision parameters is examined on the cycle performance including the exergy efficiency as well as total product unit cost. To determine the optimum working condition for the cycle, optimization is carried out by modifying all parameters to attain either the maximum exergy efficiency (EOD case) or the minimum product unit cost (COD case). Lastly, exergoeconomic parameters such as exergy

destruction rate, exergy destruction cost rate, and the exergoeconomic factor are assessed separately for each component to determine the most important components for modification in order to improve the cost-efficiency of the system.

## 2. System Description

Figure 1 illustrates the proposed integrated power and refrigeration system, consisting of a *modified Kalina cycle* and absorption refrigeration subsystems. The two subsystems are connected in such a way that they have a common condenser where the outlet flow (state 12) is divided into two separate streams. One flow goes through the KC to produce power (state 12b), while the other one runs through the ARC to generate cooling (state 12a).

Initially, the temperature and pressure of the saturated ammonia-water liquid coming from the Kalina absorber (state 1) are increased in pump1, reheating exchangers (RHE) 1 and 2, respectively before entering to the boiler (states 1 to 4). Meanwhile, the geothermal hot water is supplied to the Kalina boiler (state 32b) to provide heat for the ammonia-water solution passing through it.

The exiting vapor-liquid solution from the boiler (state 5) is divided into two segments in separator 1 (Sep 1). The separated weak solution with a lower ammonia concentration (saturated liquid) is cooled in the RHE2 (state 5b) and after an isenthalpic process in throttle valve1 (state 8), the solution passes to mixer 1 (state 8). Another solution from separator 1 (saturated vapor-state 5a) produces power in the turbine before being mixed in mixer 1 (state 6) with the other solution coming from valve 1 (state 9).

After being cooled in RHE2 (state 10), the mixed stream is combined with the ARC generator strong exiting solution (state 20) in mixer 2. Next, the divided ammonia-water vapor in separator 2 (state 11b) passes through the absorber 2 and its vapor is absorbed, while the saturated ammonia-water liquid (state 11a) is condensed and rejects its heat to the cooling water passing through the condenser (state 12).

Regarding the ARC, the separated part of the solution coming from the KC condenser is vaporized in the ARC evaporator (state 15) after being cooled in the absorption heat exchanger (AHE) and throttled in valve 2 (states 13 and 14).

The vaporized solution then enters the ARC absorber (state 16) and is mixed with the weak solution coming from the generator via the solution heat exchanger (SHE) and valve 2 (states 21 and 22). The outlet stream of the absorber1 (state 17) goes through the SHE before entering the generator (state 19).

Due to the rectifying process in the rectification column of the generator, the outlet solution has a 0.99 ammonia concentration (state 20), which is mixed with the solution coming from the turbine in the KC. Finally, after transferring heat to the generator (state 32a) and the boiler (state 32b), the geothermal water is injected to the Earth again (state 35).

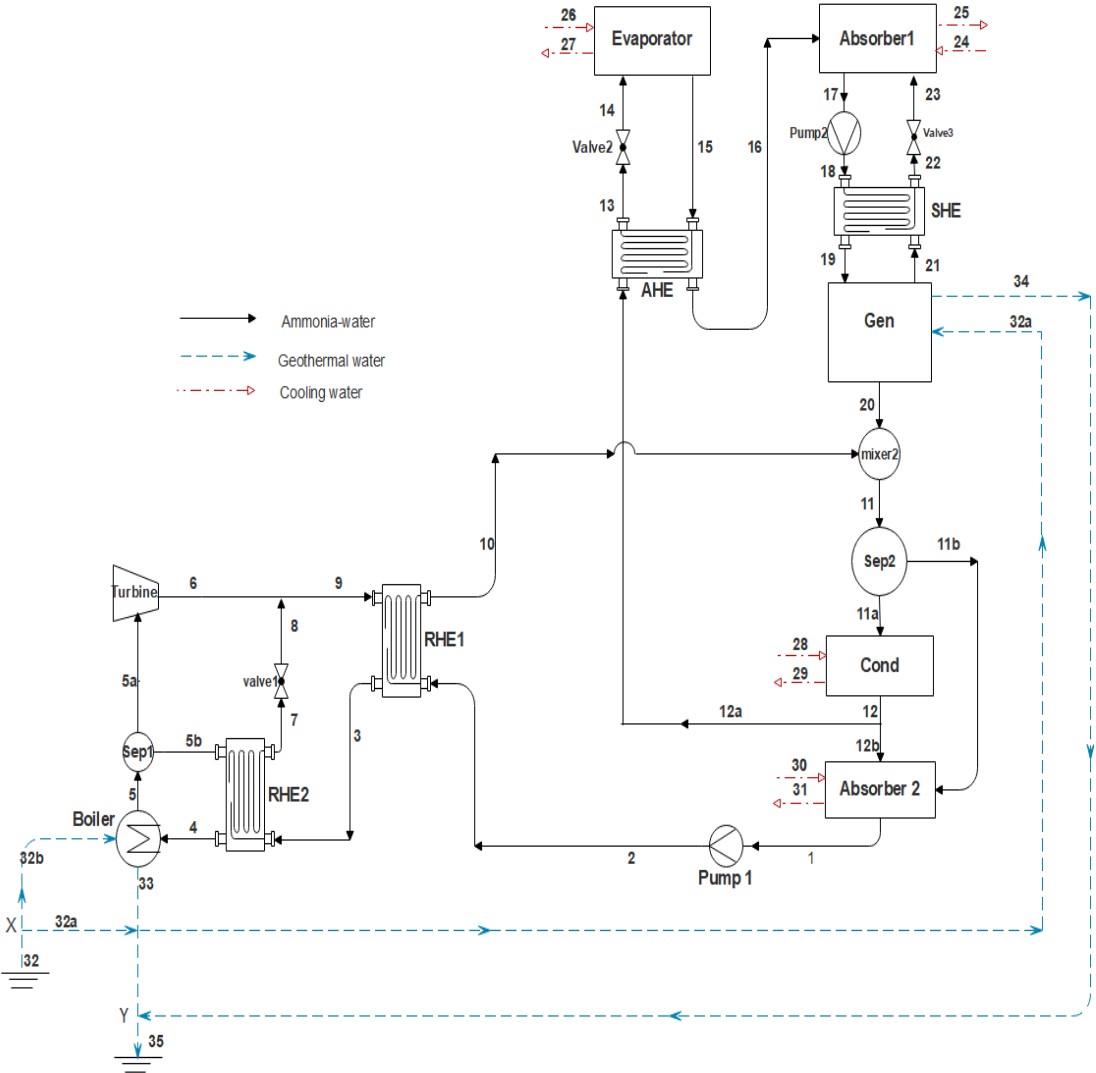

**Figure 1.** Schematic of the proposed integrated power and refrigeration cycle.

## 3. Modeling and Analyses

### 3.1. Thermodynamic Analysis

Each component in the cogeneration cycle in Figure 1 is regarded as a control volume. Initially, mass conservation relations, as well as the first and second laws of thermodynamics, are applied for each component. All modeling processes are carried out by applying the EES software developed by Ibrahim and Klein [28]. Furthermore, the following simplifying assumptions are invoked during modeling:

- The system operates at steady state;
- Pressure drops are negligible in all heat exchangers and pipes;
- The fluid flows exiting the absorbers and entering the pumps and generator (state 19) are saturated ammonia-water liquid;
- The fluids exiting the generator (state 20) and entering the absorbers are saturated ammonia-water vapor;
- Isentropic efficiencies of the pumps and turbine are constant in value;
- Exiting fluid flows from the separators are saturated liquids and vapors;
- The working fluid undergoes isenthalpic processes in the throttle valves;
- Changes in potential and kinetic energies are negligible.

The mathematical model of each component, accounting for the above assumptions, are now written [29].

The mass rate balance equation for a general component can be written as:

$$\sum \dot{m}_{in} = \sum \dot{m}_{out} \tag{1}$$

$$\sum (\dot{m}.X)_{in} = \sum (\dot{m}.X)_{out} \tag{2}$$

where $X$ denotes ammonia concentration in the ammonia-water solution.

Energy and exergy conservation equations for the entire system are given respectively as:

$$\sum \dot{m}_{in}h_{in} + \dot{Q}_{cv} - \sum \dot{m}_{out}\, h_{out} - \dot{W}_{cv} = 0 \tag{3}$$

$$\sum \dot{E}_{in} - \sum \dot{E}_{out} + \sum \dot{E}_{heat} + \sum \dot{W} - \dot{E}_{destruction} = 0 \tag{4}$$

In Equation (4), each exergy term comprises the sum of chemical and physical exergies. The variations of other kinds of exergies in this study are negligible. Hence, we can write:

$$\dot{E} = \dot{E}_{ch} + \dot{E}_{ph} \tag{5}$$

Chemical and physical exergy rates can be expressed by the following equations in the absence of electrical work, and the variations of kinetic and potential energies [29]:

$$\dot{E}_k = \dot{m}[(h - h_0) - T_0(s - s_0)] \tag{6}$$

$$\dot{E}_{ch} = \dot{m}[(\frac{X}{M_{NH_3}})e_{NH_3}^0 - T_0(\frac{1 - T_{0.}}{M_{H_2O}})e_{H_2O}^0] \tag{7}$$

where subscript $^0$ denotes the restricted dead state, which has the same temperature and pressure as the standard environment ($T_0 = 25\,°C$ and $P_0 = 1$ bar). Therefore, $h_0$ and $s_0$ are the specific enthalpy and specific entropy of the dead state, respectively. Also, $e_0$ denotes the standard chemical exergy of a substance [29].

To assess the system performance in this study, the thermal efficiency, defined as the energy utilization factor (EUF) in the cogeneration cycles, and the exergy efficiency are calculated. The EUF factor can be described as the produced energy in the cogeneration cycle, including both power and refrigeration, divided by the input energy to the cycle from the heat source [28–30]:

$$EUF = \eta_{thermal} = \frac{\dot{W}_{net} + \dot{Q}_{cooling}}{\dot{Q}_{in}} \times 100 \tag{8}$$

where $\dot{W}_{net}$ denotes the net mechanical power produced in the KC and $\dot{Q}_{cooling}$ is the refrigeration produced in the ARC evaporator. Utilizing to Figure 1, these parameters can be calculated as follows:

$$\dot{W}_{net} = \dot{W}_{turbine} - \dot{W}_{pump1} - \dot{W}_{pump2} \tag{9}$$

$$\dot{Q}_{cooling} = \dot{m}_{26}(h_{26} - h_{27}) \tag{10}$$

$$\dot{Q}_{cooling} = \dot{m}_{26}(h_{26} - h_{27}) \tag{11}$$

The exergy efficiency of the overall cycle can be written as follows [30]:

$$\eta_{exergy} = \frac{\dot{E}_{cooling} + \dot{W}_{net}}{\dot{E}_{in}} \times 100 \tag{12}$$

where

$$\dot{E}_{in} = \dot{E}_{32} - \dot{E}_{35} \tag{13}$$

$$\dot{E}_{cooling} = \dot{E}_{27} - \dot{E}_{26} \tag{14}$$

With the above equations, energy and exergy relations for each component are written (see Table 1). Stream numbers correspond to those in Figure 1.

**Table 1.** Energy and exergy relations for the proposed cycle.

| Component | Energy Relation | Exergy Relation |
|---|---|---|
| **KC** | | |
| **Pump 1** | $\dot{W}_{pump1} = \dot{m}_1(h_2 - h_1)$ | $\dot{E}_1 - \dot{E}_2 + \dot{W}_{pump1} - \dot{E}_{destruction} = 0$ |
| **RHE 2** | $\dot{m}_3 h_3 + \dot{m}_{5b} h_{5b} = \dot{m}_4 h_4 + \dot{m}_7 h_7$ | $\dot{E}_3 - \dot{E}_4 + \dot{E}_{5b} - \dot{E}_7 - \dot{E}_{destruction} = 0$ |
| **RHE 1** | $\dot{m}_2 h_2 + \dot{m}_9 h_9 = \dot{m}_3 h_3 + \dot{m}_{10} h_{10}$ | $\dot{E}_2 - \dot{E}_{10} + \dot{E}_9 - \dot{E}_3 - \dot{E}_{destruction} = 0$ |
| **Boiler** | $\dot{m}_4 h_4 + \dot{m}_{32b} h_{32b} = \dot{m}_5 h_5 + \dot{m}_{33} h_{33}$ | $\dot{E}_4 - \dot{E}_{33} + \dot{E}_{32b} - \dot{E}_5 - \dot{E}_{destruction} = 0$ |
| **Separator 1** | $\dot{m}_5 h_5 = \dot{m}_{5a} h_{5a} + \dot{m}_{5b} h_{5b}$<br>$\dot{m}_5 = \dot{m}_{5a} + \dot{m}_{5b}$ | $\dot{E}_5 - \dot{E}_{5a} - \dot{E}_{5b} - \dot{E}_{destruction} = 0$ |
| **Turbine** | $\dot{m}_{5a} h_{5a} = \dot{m}_6 h_6 + \dot{W}_{turbine}$ | $\dot{E}_{5a} - \dot{E}_6 + \dot{W}_{turbine} - \dot{E}_{destruction} = 0$ |
| **Mixer 1** | $\dot{m}_6 h_6 + \dot{m}_8 h_8 = \dot{m}_9 h_9$<br>$\dot{m}_6 + \dot{m}_8 = \dot{m}_9$ | $\dot{E}_6 + \dot{E}_8 - \dot{E}_9 - \dot{E}_{destruction} = 0$ |
| **Mixer 2** | $\dot{m}_{10} h_{10} + \dot{m}_{20} h_{20} = \dot{m}_{11} h_{11}$<br>$\dot{m}_{10} + \dot{m}_{20} = \dot{m}_{11}$ | $\dot{E}_{10} + \dot{E}_{20} - \dot{E}_{11} - \dot{E}_{destruction} = 0$ |
| **Separator 2** | $\dot{m}_{11a} h_{11a} + \dot{m}_{11b} h_{11b} = \dot{m}_{11} h_{11}$<br>$\dot{m}_{11a} + \dot{m}_{11b} = \dot{m}_{11}$ | $\dot{E}_{11} - \dot{E}_{11a} - \dot{E}_{11b} - \dot{E}_{destruction} = 0$ |
| **Condenser** | $\dot{m}_{11a} h_{11a} + \dot{m}_{28} h_{28} = \dot{m}_{29} h_{29} + \dot{m}_{12} h_{12}$ | $\dot{E}_{11a} - \dot{E}_{12} + \dot{E}_{28} - \dot{E}_{29} - \dot{E}_{destruction} = 0$ |
| **Absorber 2** | $\dot{m}_{12b} h_{12b} + \dot{m}_{11b} h_{11b} + \dot{m}_{30} h_{30} = \dot{m}_{31} h_{31} + \dot{m}_1 h_1$ | $\dot{E}_{11b} + \dot{E}_{12b} + \dot{E}_{30} - \dot{E}_1 - \dot{E}_{31} - \dot{E}_{destruction} = 0$ |
| **Spread point** | $\dot{m}_{12b} h_{12b} + \dot{m}_{12a} h_{12a} = \dot{m}_{12} h_{12}$<br>$\dot{m}_{12a} + \dot{m}_{12b} = \dot{m}_{12}$ | $\dot{E}_{12} - \dot{E}_{12a} - \dot{E}_{12b} - \dot{E}_{destruction} = 0$ |
| **ARC** | | |
| **AHE** | $\dot{m}_{12a} h_{12a} + \dot{m}_{13} h_{13} = \dot{m}_{15} h_{15} + \dot{m}_{16} h_{16}$ | $\dot{E}_{12a} - \dot{E}_{13} + \dot{E}_{15} - \dot{E}_{16} - \dot{E}_{destrcution} = 0$ |
| **Evaporator** | $\dot{m}_{26} h_{26} + \dot{m}_{14} h_{14} = \dot{m}_{27} h_{27} + \dot{m}_{15} h_{15}$ | $\dot{E}_{14} - \dot{E}_{15} + \dot{E}_{26} - \dot{E}_{27} - \dot{E}_{destruction} = 0$ |
| **Absorber 1** | $\dot{m}_{16} h_{16} + \dot{m}_{24} h_{24} + \dot{m}_{23} h_{23} = \dot{m}_{17} h_{17} + \dot{m}_{25} h_{25}$ | $\dot{E}_{16} + \dot{E}_{23} + \dot{E}_{24} - \dot{E}_{17} - \dot{E}_{25} - \dot{E}_{destruction} = 0$ |
| **SHE** | $\dot{m}_{18} h_{18} + \dot{m}_{21} h_{21} = \dot{m}_{19} h_{19} + \dot{m}_{22} h_{22}$ | $\dot{E}_{18} + \dot{E}_{21} - \dot{E}_{22} - \dot{E}_{19} - \dot{E}_{destruction} = 0$ |
| **Pump 2** | $\dot{W}_{pump2} = \dot{m}_{17}(h_{18} - h_{17})$ | $\dot{E}_{17} - \dot{E}_{18} + \dot{W}_{pump2} - \dot{E}_{destruction} = 0$ |
| **Generator** | $\dot{m}_{19} h_{19} + \dot{m}_{32a} h_{32a} = \dot{m}_{21} h_{21} + \dot{m}_{34} h_{34} + \dot{m}_{20} h_{20}$ | $\dot{E}_{19} + \dot{E}_{32a} - \dot{E}_{21} - \dot{E}_{34} - \dot{E}_{20} - \dot{E}_{destruction} = 0$ |
| **Geothermal** | | |
| **Point X** | $\dot{m}_{32} h_{32} = \dot{m}_{32a} h_{32a} + \dot{m}_{32b} h_{32b}$ | $\dot{E}_{32} - \dot{E}_{32a} - \dot{E}_{32b} - \dot{E}_{destruction} = 0$ |
| **Point Y** | $\dot{m}_{34} h_{34} + \dot{m}_{33} h_{33} = \dot{m}_{35} h_{35}$ | $\dot{E}_{33} + \dot{E}_{34} - \dot{E}_{35} - \dot{E}_{destruction} = 0$ |

*3.2. Thermoeconomic Analysis*

Thermoeconomics is a relatively new branch of engineering which combines exergy and economic analyses in order to examine systems better and to design more energy and cost-efficient systems. This approach provides worthwhile information about cost formation process and unit exergy cost for each stream. To accomplish this, the cost balance equation along with auxiliary equations need to be written [28,29]:

$$\sum \dot{C}_{e,k} + \dot{C}_{w,k} = \sum \dot{C}_{i,k} + \dot{C}_{q,k} + \dot{Z}_k \tag{15}$$

where

$$\dot{C}_j = \dot{c}_j \dot{E}_j \tag{16}$$

In the above equations, $\dot{C}$ and $c$ represent cost rate of exergy ($\$/\text{hr.}$) and cost per unit exergy ($\$/\text{GJ}$) respectively. Meanwhile, $\dot{C}_w$ and $\dot{C}_q$ refer respectively to the costs associated with work and heat transfer rates in each component. From Equation (15), it can be seen that the sum of costs due to outlet exergy flow rates and producing power in a component, is equal to the sum of all investment costs plus entering exergy flow rates to the component. The investment cost ($\dot{Z}_k$) for a component can be defined as the sum of capital investment ($\dot{Z}_k^{CI}$) and operating and maintenance cost rates ($\dot{Z}_k^{OM}$):

$$\dot{Z}_k = \dot{Z}_k^{CI} + \dot{Z}_k^{OM} \tag{17}$$

The levelized capital investment cost rate can be written by the following equation [28]:

$$\dot{Z}_k = (\frac{CRF}{\tau})Z_k + (\frac{\gamma_k}{\tau})Z_k + \omega_k \dot{E}_{p,k} + \frac{R_k}{\tau} \tag{18}$$

In this study, $Z_k$ is calculated from functions in the appendix.

In Equation (18), $\tau$ denotes annual operation hours for a plant, and $\gamma_k$ and $\omega_k$ are fixed and variable operating costs respectively. Also, $R_k$ refers to other costs which are separate from operation and maintenance costs. The first term in Equation (18) is much larger than the two last terms, which can therefore be neglected. Finally, CRF is the capital recovery factor which can be written as follows [28,29]:

$$CRF = \frac{i_r(1 + i_r)^n}{(1 + i_r)^n - 1} \tag{19}$$

where $i_r$ is the annual interest rate and n is the plant useful operation life.

The overall production cost rate and the total product unit cost for the entire system can be expressed respectively as follows [24]:

$$\dot{C}_{p,total} = \frac{\dot{C}_{p,cooling} + \dot{C}_{p,power}}{\dot{W}_{net} + \dot{E}_{cooling}} \tag{20}$$

$$c_{p,total} = c_{p,cooling} + c_{p,power} \tag{21}$$

For each component in Figure 1 the cost balance, as well as auxiliary cost equations, are listed in Table 2.

**Table 2.** Cost rate balance and auxiliary equations for each component of the system.

| Component | Cost Rate Balance | Auxiliary Relations |
|---|---|---|
| **KC** | | |
| **Pump1** | $\dot{C}_1 + \dot{Z}_{pump1} + \dot{C}_{w,pump1} = \dot{C}_2$ | |
| **RHE 2** | $\dot{C}_9 + \dot{C}_2 + \dot{Z}_{RHE2} = \dot{C}_3 + \dot{C}_{10}$ | $c_9 = c_{10}$ |
| **RHE1** | $\dot{C}_3 + \dot{Z}_{RHE1} + \dot{C}_{5b} = \dot{C}_4 + \dot{C}_7$ | $c_{5b} = c_7$ |
| **Boiler** | $\dot{C}_4 + \dot{Z}_{boiler} + \dot{C}_{32b} = \dot{C}_5 + \dot{C}_{33}$ | $c_{32b} = c_{33}$ |
| **Separator 1** | $\dot{C}_5 = \dot{C}_{5a} + \dot{C}_{5b}$ | $c_{5a} = c_{5b}$ |
| **Turbine** | $\dot{C}_5 + \dot{Z}_{turbine} = \dot{C}_6 + \dot{C}_{w,turbine}$ | $c_{5a} = c_6$, $c_{w,turbine} = c_{w,pump2}$ |

| | | |
|---|---|---|
| **Valve 1** | $\dot{C}_7 = \dot{C}_8$ | |
| **Mixer 1** | $\dot{C}_6 + \dot{C}_8 = \dot{C}_9$ | |
| **Mixer 2** | $\dot{C}_{10} + \dot{C}_{20} = \dot{C}_{11}$ | |
| **Separator 2** | $\dot{C}_1 = \dot{C}_{11a} + \dot{C}_{11b}$ | $c_{11a} = c_{11b}$ |
| **Condenser** | $\dot{C}_{11a} + \dot{Z}_{condesner} + \dot{C}_{28} = \dot{C}_{12} + \dot{C}_{29}$ | $c_{11a} = c_{12}$, $c_{28} = 0$ |
| **Absorber 2** | $\dot{C}_{12b} + \dot{Z}_{absorber2} + \dot{C}_{11b} + \dot{C}_{30} = \dot{C}_1 + \dot{C}_{31}$ | $\dfrac{\dot{C}_{12b} + \dot{C}_{11b}}{\dot{E}_{12b} + \dot{E}_{11b}} = \dfrac{\dot{C}_1}{\dot{E}_1}$, $c_{30} = 0$ |
| **Spread point** | $\dot{C}_{12} = \dot{C}_{12a} + \dot{C}_{12b}$ | $c_{12a} = c_{12b}$ |

**ARC**

| | | |
|---|---|---|
| **AHE** | $\dot{C}_{12a} + \dot{Z}_{AHE} + \dot{C}_{15} = \dot{C}_{16} + \dot{C}_{13}$ | $c_{12a} = c_{13}$ |
| **Valve 2** | $\dot{C}_{13} = \dot{C}_{14}$ | |
| **Evaporator** | $\dot{C}_{14} + \dot{Z}_{evaporator} + \dot{C}_{26} = \dot{C}_{15} + \dot{C}_{27}$ | $c_{14} = c_{15}$, $c_{26} = 0$ |
| **Absorber 1** | $\dot{C}_{16} + \dot{Z}_{absorber1} + \dot{C}_{23} + \dot{C}_{24} = \dot{C}_{17} + \dot{C}_{25}$ | $\dfrac{\dot{C}_{16} + \dot{C}_{23}}{\dot{E}_{16} + \dot{E}_{23}} = \dfrac{\dot{C}_{17}}{\dot{E}_{17}}$, $c_{24} = 0$ |
| **SHE** | $\dot{C}_{18} + \dot{Z}_{SHE} + \dot{C}_{21} = \dot{C}_{19} + \dot{C}_{22}$ | $c_{18} = c_{19}$ |
| **Pump 2** | $\dot{C}_{17} + \dot{Z}_{pump2} + \dot{C}_{w,pump2} = \dot{C}_{18}$ | $c_{w,pump1} = c_{w,pump2}$ |
| **Valve 3** | $\dot{C}_{22} = \dot{C}_{23}$ | |
| **Generator** | $\dot{C}_{19} + \dot{Z}_{generator} + \dot{C}_{32a} = \dot{C}_{21} + \dot{C}_{34} + \dot{C}_{20}$ | $\dfrac{\dot{C}_{20} - \dot{C}_{19}}{\dot{E}_{20} - \dot{E}_{19}} = \dfrac{\dot{C}_{21} - \dot{C}_{19}}{\dot{E}_{21} - \dot{E}_{19}}$, $c_{34} = c_{32a}$ |

**Geothermal**

| | | |
|---|---|---|
| **Point X** | $\dot{C}_{32} = \dot{C}_{32a} + \dot{C}_{32b}$ | $c_{32a} = c_{32b}$, $c_{32} = \text{known}$ |
| **Point Y** | $\dot{C}_{33} + \dot{C}_{34} = \dot{C}_{35}$ | |

Next, the exergoeconomic factor ($f_k$), which plays a key role in evaluating the economic performance of the system and components, is introduced. This factor can be defined as the ratio of the costs associated with the operating and maintenance plus capital investment cost rates ( $\dot{Z}_k = \dot{Z}_k^{CI} + \dot{Z}_k^{OM}$ ), namely non-exergy related costs, to the costs associated with exergy destruction plus non-exergy related cost rates. That is,

$$f_k = \frac{\dot{Z}_k}{\dot{Z}_k + c_{F,k}(\dot{E}_{D,k} + \dot{E}_{L,k})} \tag{22}$$

A high value of this factor, which lies between zero and one, implies for a component that if investment cost declines, the component would be more cost-efficient. Conversely, a low value of the factor indicates that cost savings could be achieved by decreasing the exergy destruction cost of the component.

Another important parameter is the cost of exergy destruction rate. This parameter does not appear in the cost balance equation (Equation (15)). It can be referred as a hidden cost which is associated with the exergy destruction rate and can be calculated for each component as follows [25]:

$$\dot{C}_{D,k} = c_{F,k} \times \dot{E}_{D,k} \tag{23}$$

where $c_{F,k}$ is the average unit cost of fuel for each system component.

## 4. Results

First, the two subsystems introduced in the proposed cogeneration cycle in Figure 1 (KC and ARC) are validated individually referring to references [31] and [32] respectively. For the proposed modified KC in this study, as there is no data reported in literature, the data for a similar configuration [31] i.e., for the KC-11, are used to validate the results obtained in the present work for this cycle. The comparison is shown in Figure 2.

Also, Table 3 compares pressure, temperature, ammonia concentration, and ammonia mass flow rate obtained for the ARC in the present study with those reported in reference [32].

It can be seen that both of our results are in good agreements with those reported in literature, with a maximum relative difference of 0.5 percent.

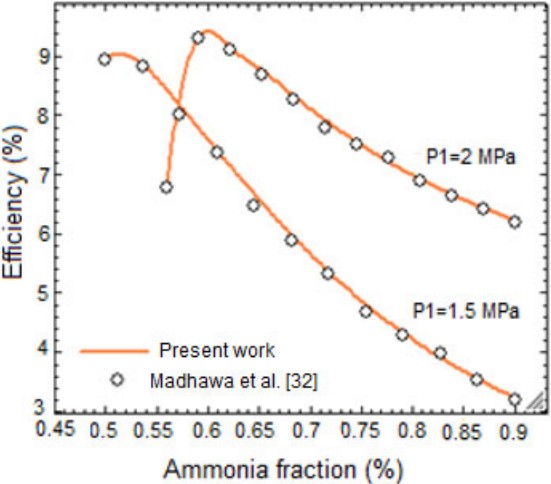

**Figure 2.** Comparison of the results obtained in the present work with those reported for the KC in reference [31].

**Table 3.** Comparison of the results obtained in the present work with those reported for the ARC in reference [32].

| State | Pressure (bar) | | Temperature (K) | | Ammonia Concentration (%) | | Mass Flow Rate (lbm/min) | |
|---|---|---|---|---|---|---|---|---|
| | Ref [33] | Present Study | Ref [33] | Present Study | Ref [33] | Present Study | Ref [33] | Present Study |
| 1 | 2.06 | 2.06 | 299.3 | 301.2 | 0.408 | 0.411 | 1.98 | 2.05 |
| 2 | 13.7 | 13.7 | 299.4 | 301.3 | 0.408 | 0.411 | 1.98 | 2.05 |
| 3 | 13.7 | 13.7 | 366.4 | 366.5 | 0.408 | 0.411 | 1.98 | 2.05 |
| 4 | 13.7 | 13.7 | 388.7 | 388.6 | 0.298 | 0.304 | 1.67 | 1.752 |
| 5 | 13.7 | 13.7 | 309.2 | 311.3 | 0.298 | 0.304 | 1.67 | 1.75 |
| 6 | 2.06 | 2.06 | 309.3 | 311.4 | 0.298 | 0.304 | 1.67 | 1.75 |
| 7 | 13.7 | 13.7 | 327.5 | 327.6 | 0.996 | 0.997 | 0.312 | 0.309 |
| 8 | 13.7 | 13.7 | 309.2 | 308.9 | 0.996 | 0.997 | 0.312 | 0.309 |
| 9 | 13.7 | 13.7 | 303.1 | 303.2 | 0.996 | 0.997 | 0.312 | 0.309 |
| 10 | 2.06 | 2.06 | 255.3 | 255.1 | 0.996 | 0.997 | 0.312 | 0.309 |
| 11 | 2.06 | 2.06 | 279.6 | 277.3 | 0.996 | 0.997 | 0.312 | 0.309 |
| 12 | 2.06 | 2.06 | 287 | 287.1 | 0.996 | 0.997 | 0.312 | 0.309 |

The values of the initial input data for the proposed cogeneration system are listed in Table 4. It is assumed that the system operates 8000 h a year producing power and cooling [24].

**Table 4.** Input parameters for the proposed cogeneration cycle.

| Parameter | Symbol | Value |
|---|---|---|
| Environment temperature | $T_0$ | 25°C |
| Environment pressure | $P_0$ | 1 bar |
| Geothermal inlet temperature | $T_{32}$ | 130°C |
| Geothermal inlet mass flow rate | $m_{32}$ | $100 \; \frac{kg}{s}$ |
| Geothermal reinjection temperature | $T_{35}$ | 90°C |
| Pinch point temperature difference | $\Delta T_{pp}$ | 8°C |
| Pumps isentropic efficiency | $\eta_{pump}$ | 0.8 |
| Turbine isentropic efficiency | $\eta_{turbine}$ | 0.85 |
| Evaporator pressure | $P_{evaporator}$ | 1.6 bar |
| Condenser pressure | $P_{condesner}$ | 12 bar |
| Boiler pressure | $P_{boiler}$ | 24 bar |
| Cooling water temperature | $T_{cw}$ | 25°C |
| Exiting water temperature | $T_{ew}$ | 35°C |
| Generator inlet temperature | $T_{19}$ | 94°C |
| Generator outlet temperature | $T_{20}$ | 51°C |
| Generator outlet temperature | $T_{21}$ | 116°C |
| Heat exchanger temperature drop | $\delta_{HE}$ | 10°C |
| Annual operation hours | $\tau$ | $8000 \; \frac{hr.}{year}$ |
| Interest rate | $i_r$ | 15% |
| Plant operation life | $n$ | 20 year |
| Temperature of evaporator inlet water | $T_{26}$ | 15°C |
| Temperature of evaporator outlet water | $T_{27}$ | 10°C |

## 5. Performance Results

By applying the equations provided in the previous sections as well as the values in Table 4, values are determined for pressure, temperature, ammonia mass flow rate, ammonia concentration, exergy rate, unit exergy cost rate, and exergy cost rate. The results are presented in Table 5. It can be seen that the unit exergy cost for absorber and evaporator are the highest among the others, respectively. This is mainly because of the loss of heat in absorber and evaporator.

**Table 5.** Thermodynamic analysis results for the cogeneration cycle in Figure 1.

| State | Working Fluid | Pressure (bar) | Temperature $(K)$ | Mass Flow Rate $\dot{m}_i \; (\frac{kg}{s})$ | Exergy Rate $(MW)$ | Unit Exergy Cost $c_i \; (\frac{\$}{GJ})$ | Exergy Cost Rate $\dot{C}_i \; (\frac{\$}{hr.})$ | Ammonia Concentration (%) |
|---|---|---|---|---|---|---|---|---|
| 1 | Ammonia-water | 12 | 306.5 | 14.7 | 272.04 | 4.72 | 4630 | 0.92 |
| 2 | Ammonia-water | 24 | 306.9 | 14.7 | 272.07 | 4.72 | 4629 | 0.92 |
| 3 | Ammonia-water | 24 | 317.7 | 14.7 | 272.10 | 4.73 | 4633 | 0.92 |

| | | | | | | | |
|---|---|---|---|---|---|---|---|
| 4 | Ammonia-water | 24 | 322.7 | 14.7 | 272.13 | 4.73 | 4636 | 0.92 |
| 5 | Ammonia-water | 24 | 360.9 | 14.7 | 273.96 | 4.71 | 4653 | 0.92 |
| 5a | Ammonia-water | 24 | 360.9 | 12.3 | 247.31 | 4.71 | 4201 | 0.99 |
| 5b | Ammonia-water | 24 | 360.9 | 2.34 | 266.47 | 4.71 | 452.6 | 0.59 |
| 6 | Ammonia-water | 12 | 323.9 | 12.3 | 246.10 | 4.71 | 4180 | 0.99 |
| 7 | Ammonia-water | 24 | 327.7 | 2.34 | 265.97 | 4.71 | 451.8 | 0.59 |
| 9 | Ammonia-water | 12 | 327.7 | 14.7 | 272.70 | 4.71 | 4632 | 0.92 |
| 10 | Ammonia-water | 12 | 322.4 | 14.7 | 272.63 | 4.71 | 4631 | 0.92 |
| 11 | Ammonia-water | 12 | 322.4 | 15.6 | 291.3 | 4.71 | 4958 | 0.93 |
| 11a | Ammonia-water | 12 | 322.4 | 12.5 | 250.88 | 4.72 | 4270 | 0.99 |
| 11b | Ammonia-water | 12 | 322.4 | 3.13 | 40.401 | 4.72 | 687.6 | 0.66 |
| 12 | Ammonia-water | 12 | 306.2 | 12.5 | 250.82 | 4.72 | 4269 | 0.99 |
| 12a | Ammonia-water | 12 | 306.2 | 0.93 | 18.67 | 4.72 | 317.9 | 0.99 |
| 12b | Ammonia-water | 12 | 306.2 | 11.5 | 232.15 | 4.72 | 3951 | 0.99 |
| 13 | Ammonia-water | 12 | 301.2 | 0.93 | 18.650 | 4.72 | 317.5 | 0.99 |
| 14 | Ammonia-water | 1.6 | 249.4 | 0.93 | 18.620 | 4.73 | 317.5 | 0.99 |
| 15 | Ammonia-water | 1.6 | 249.4 | 0.93 | 18.620 | 4.73 | 317.5 | 0.99 |
| 16 | Ammonia-water | 1.6 | 281.4 | 0.93 | 18.240 | 4.82 | 320.1 | 0.99 |
| 17 | Ammonia-water | 1.6 | 300.0 | 6.74 | 47.422 | 4.88 | 833.9 | 0.38 |
| 18 | Ammonia-water | 12 | 300.2 | 6.74 | 47.432 | 4.88 | 833.8 | 0.38 |
| 19 | Ammonia-water | 12 | 367.2 | 6.74 | 47.640 | 4.88 | 837.5 | 0.38 |
| 20 | Ammonia-water | 12 | 324.2 | 0.93 | 18.670 | 4.86 | 326.8 | 0.99 |
| 21 | Ammonia-water | 12 | 389.2 | 5.80 | 29.360 | 4.87 | 515.2 | 0.28 |
| 22 | Ammonia-water | 12 | 310.4 | 5.81 | 29.072 | 4.92 | 514.9 | 0.28 |
| 23 | Ammonia-water | 1.6 | 310.6 | 5.81 | 29.061 | 4.92 | 514.9 | 0.28 |
| 24 | Water | 1 | 298.2 | 43.0 | 0.0 | 0.0 | 0.0 | 0.0 |
| 25 | Water | 1 | 308.2 | 43.0 | 29.660 | 14.0 | 1.49 | |
| 26 | Water | 1 | 288.2 | 1.84 | 0.0100 | 0.0 | 0.0 | |
| 27 | Water | 1 | 283.2 | 1.84 | 0.0100 | 244.5 | 2.65 | |
| 28 | Water | 1 | 298.2 | 24.0 | 0.0 | 0.0 | 0.0 | |
| 29 | Water | 1 | 308.2 | 0.01 | 65.74 | 3.92 | 3.92 | |
| 30 | Water | 1 | 298.2 | 315.2 | 0 | 0 | 0 | |
| 31 | Water | 1 | 308.2 | 315.2 | 0.21 | 5952 | 4651 | |
| 32 | Water | 2.7 | 403.2 | 100 | 6.37 | 1.37 | 31.48 | |
| 32a | Water | 2.7 | 403.2 | 45.4 | 2.89 | 1.37 | 14.31 | |
| 32b | Water | 2.7 | 403.2 | 54.5 | 3.47 | 1.37 | 17.17 | |
| 33 | Water | 2.7 | 337.5 | 54.55 | 0.54 | 1.37 | 2.69 | |
| 34 | Water | 2.7 | 393.7 | 45.45 | 2.43 | 1.37 | 12.02 | |
| 35 | Water | 2.7 | 363.2 | 100 | 2.60 | 1.57 | 14.72 | |

Table 6 summarizes the values obtained for the cogeneration cycle performance parameters, including the thermal and exergy efficiencies as well as unit costs of the power, cooling and total production. It is observed that the power unit cost is considerably higher than the cooling unit cost. This is attributed to the values of $\dot{C}_{D,k} + \dot{Z}_k$ for the components involved in power generation especially for the turbine compared to the components serving for cooling production, as indicated in Table 6.

**Table 6.** Performance results for the cogeneration cycle.

| Parameter | | Value |
|---|---|---|
| Exergy efficiency | $\eta_{exergy}$ | 26.4% |
| Thermal efficiency | $EUF = \eta_{thermal}$ | 7.10% |
| Power unit cost | $c_{power}$ | $13.7\ \$/\text{GJ}$ |
| Cooling unit cost | $c_{cooling}$ | $4.70\ \$/\text{GJ}$ |
| Total product unit cost | $c_{overall}$ | $18.4\ \$/\text{GJ}$ |

## 6. Parametric Study

In order to determine an optimum working condition, the variation of various parameters such as the condenser mass flow split ratio ($r_m$), turbine inlet pressure (TIP), and evaporator and condenser pressures, are investigated to evaluate their effects on the cycle performance, in terms of such parameters as the exergy efficiency and total product unit cost. In the following subsections, only the specified parameter is considered as variable and others are kept constant.

### 6.1. Variation of Turbine Inlet Pressure (TIP)

The variation of the total product unit cost ($c_{p,total}$) with TIP for the cogeneration cycle is depicted in Figure 3 for three values of evaporator pressure. It can be seen that as TIP rises to a certain value, $c_{p,total}$ reaches a minimum at each evaporator pressure and then rises as TIP rises further. Furthermore, it is evident that, as evaporator pressure increases, $c_{p,total}$ declines, which means that the proposed system is more cost efficient at a higher evaporator pressure. The trend shown in Figure 3 can be explained by the results in Figure 4, considering that $c_{p,total}$ is the sum of $c_{p,power}$ and $c_{p,cooling}$ as indicated by Equation (21).

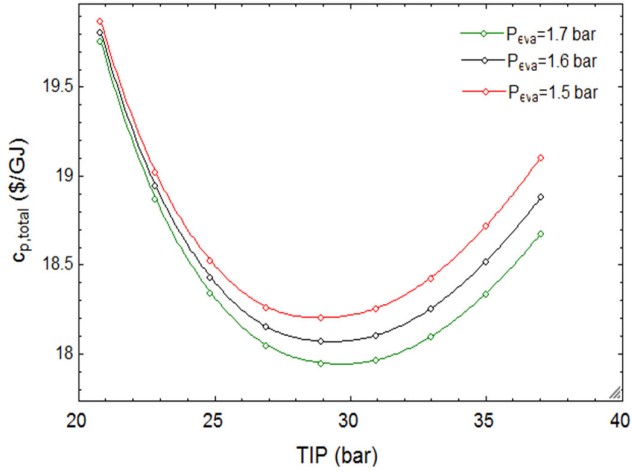

**Figure 3.** Effect of varying turbine inlet pressure on total product unit cost for several evaporator pressures.

Figure 4 shows that there are two separate optimal points for $c_{p,cooling}$ and $c_{p,power}$. The former reaches its minimum value at a TIP of 25 bar and the latter at 32 bar. Additionally, it is seen that the value of $c_{p,power}$ is considerably higher than $c_{p,cooling}$.

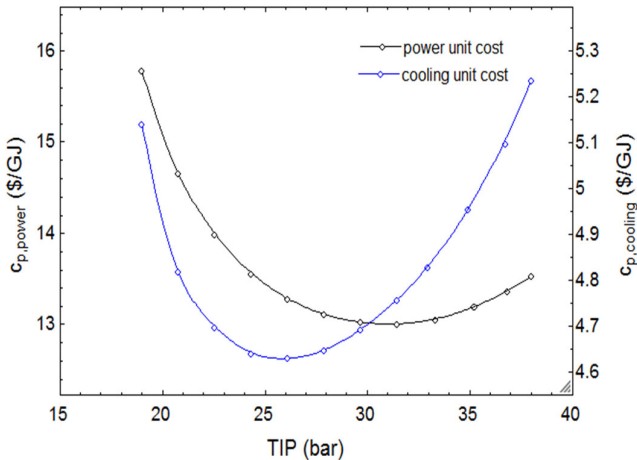

**Figure 4.** Effect of varying turbine inlet pressure on power and cooling unit costs.

Figures 5 and 6 illustrate the effect of TIP on $c_{p,total}$ as well as $\eta_{exergy}$ for various condenser pressures. While there is a minimum value for $c_{p,total}$ at each condenser pressure, $\eta_{exergy}$ is maximized at a specific value of TIP. This TIP value is higher at higher values condenser pressure. It is noted in Figures 5 and 6 that the cycle is more exergy and cost-efficient at lower condenser pressures, which means that as condenser pressure declines, $c_{p,total}$ decreases, whereas $\eta_{exergy}$ increases.

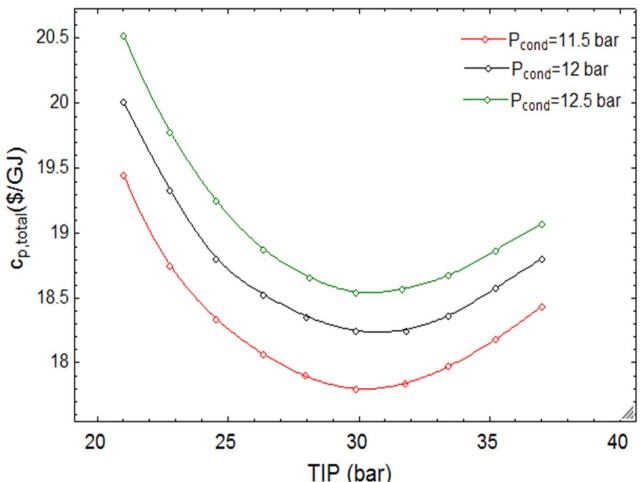

**Figure 5.** Effect of varying turbine inlet pressure on total product unit cost for several condenser pressures.

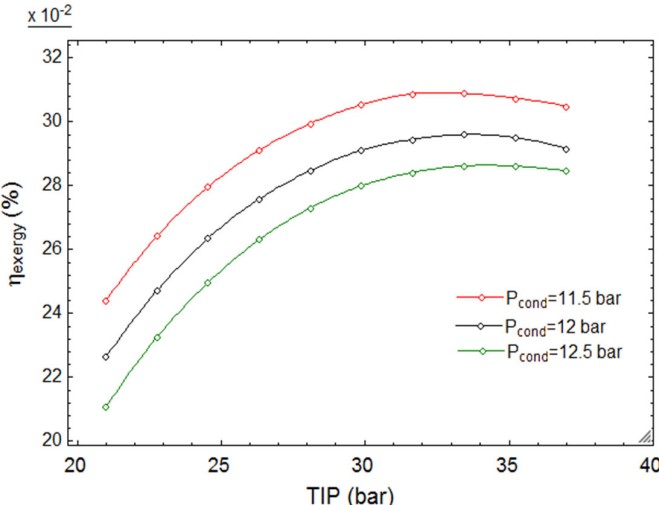

**Figure 6.** Effect of varying turbine inlet pressure on exergy efficiency for several condenser pressures.

It is noted in Figure 5 that $c_{p,total}$ increases with increasing condenser pressure, while it decreases with increasing evaporator pressure, as illustrated in Figure 3.

Variations in the thermal and exergy efficiencies as turbine inlet pressure varies are illustrated in Figure 7. While the value of thermal efficiency rises consistently with increasing TIP, $\eta_{exergy}$ reaches a peak amount at a TIP of 34 bar and then decreases. Also, it can be seen that the value of $\eta_{exergy}$ is considerably higher than the thermal efficiency in the considered interval for TIP.

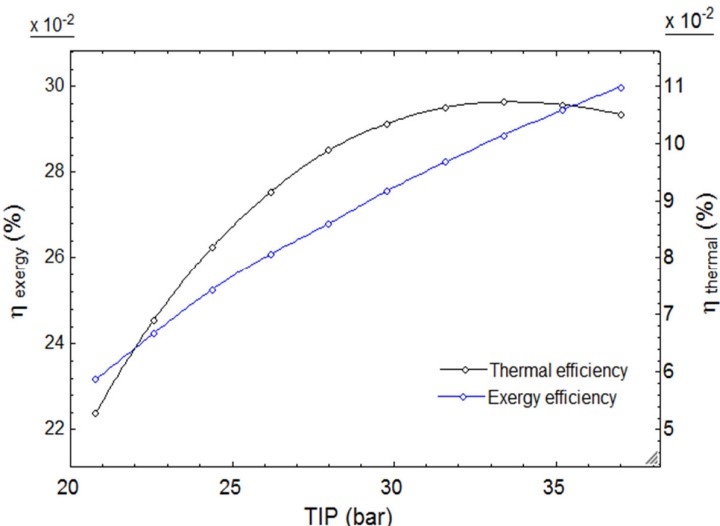

**Figure 7.** Effect of varying turbine inlet pressure on thermal and exergy efficiencies.

The justification for the variation of $\eta_{exergy}$ in Figure 7 can be explained by referring to Equation (12). For a constant value of input exergy rate ($\dot{E}_{in}$), the variation of $\dot{W}_{net}$ and $\dot{E}_{cooling}$, as indicated in Figure 8, justifies the trend for $\eta_{exergy}$.

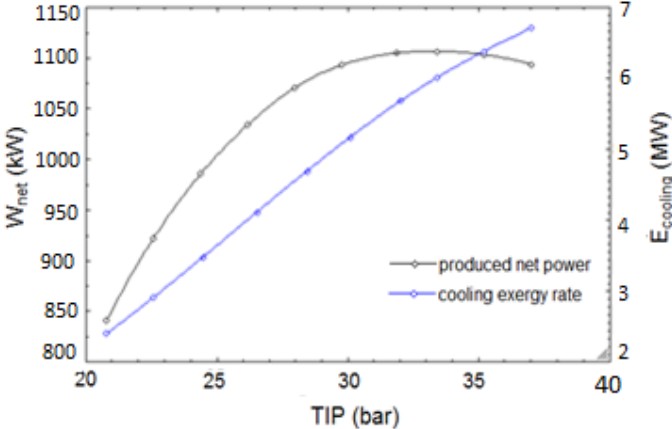

**Figure 8.** Effect of varying turbine inlet pressure on produced net power and cooling exergy rate.

### 6.2. Variation of Evaporator Pressure ($P_{eva}$)

In this section, the effect is studied of varying evaporator pressure on such performance criteria as $c_{p,total}$, $\eta_{thermal}$, and $\eta_{exergy}$. Figure 9 shows the variation of $c_{p,total}$ while evaporator pressure changes. For each value for TIP, $c_{p,total}$ takes on a minimum value at a certain value of evaporator pressure. In addition, $c_{p,total}$ decreases with increasing TIP, as shown in Figure 9. Apparently, the optimum cost of production is for an evaporator pressure between 5 and 6 bar. Also, the variations of $c_{p,cooling}$ and $c_{p,power}$ with evaporator pressure are depicted in Figure 10.

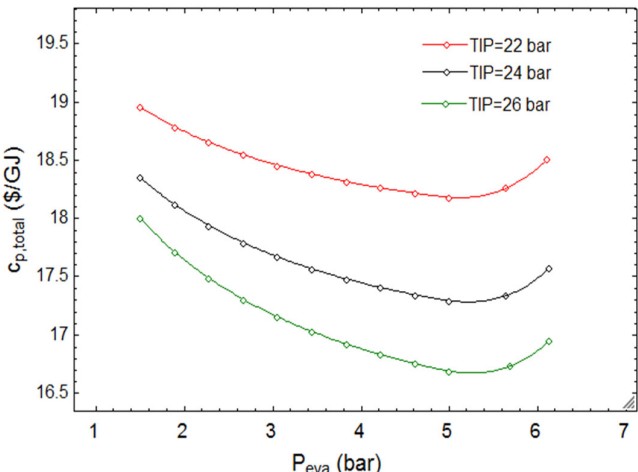

**Figure 9.** Effect of varying evaporator pressure on total product unit cost for several turbine pressures.

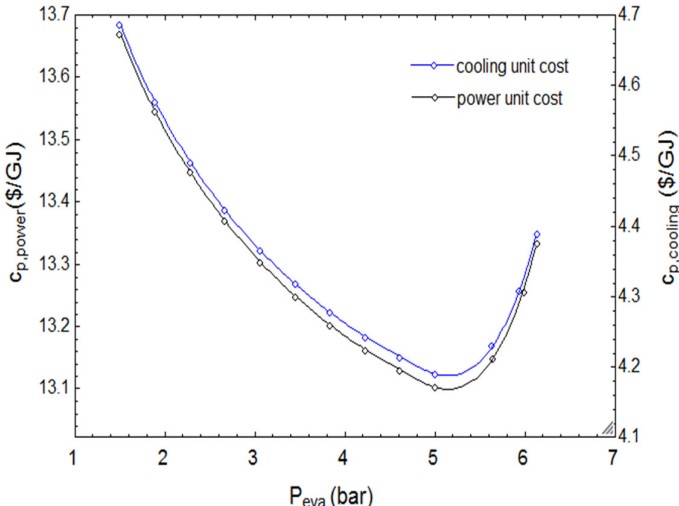

**Figure 10.** Effect of varying evaporator pressure on power and cooling unit costs.

The variation of $c_{p,total}$ with evaporator pressure is shown in Figure 11 for several condenser pressures. It can be seen that a lower condenser pressure is more cost efficient. Furthermore, there is a minimum value for $c_{p,total}$ between evaporator pressures of 5 and 6 bar.

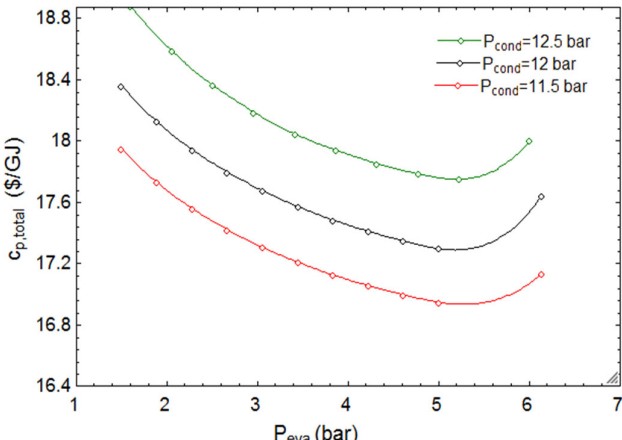

**Figure 11.** Effect of varying evaporator pressure on total product unit cost in several condenser pressures.

Figures 9 and 11 are now compared. It is evident that $c_{p,total}$ increases with increasing condenser pressure (Figure 11). This is opposed to Figure 9, where there is a reverse correlation between $c_{p,total}$ and TIP. In Figure 12, the variation of thermal and exergy efficiencies are depicted for the original case. It is evident that the value of $\eta_{exergy}$ is higher than that of $\eta_{thermal}$.

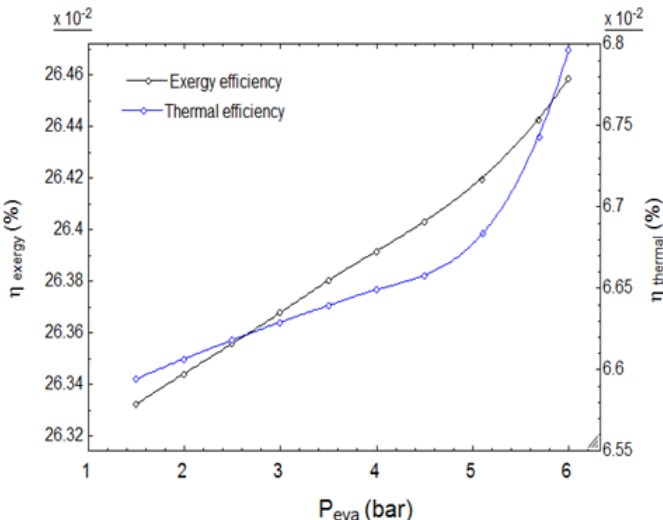

**Figure 12.** Effect of varying evaporator pressure on thermal and exergy efficiencies.

Similar to the explanation provided above for justifying the variation in $\eta_{exergy}$ (Figure 7), the justification for the variation of $\eta_{exergy}$, which is depicted in Figure 13, can be done by considering Equation (12).

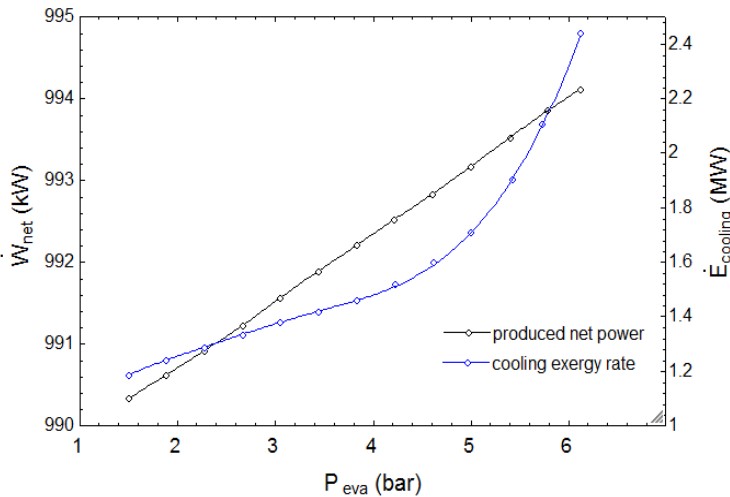

**Figure 13.** Effect of varying evaporator pressure on produced net power and cooling exergy rate.

*6.3. Variation of Condenser Mass Flow Split Ratio ($r_m$)*

The condenser mass flow split ratio is defined as the ratio of mass flow rate entering the ARC ($\dot{m}_{12a}$) to the condenser exiting mass flow rate ($\dot{m}_{12}$):

$$r_m = \frac{\dot{m}_{12a}}{\dot{m}_{12}} \tag{24}$$

The effect of varying $r_m$ on $c_{p,total}$ is illustrated for three evaporator pressures in Figure 14. For each evaporator pressure, $c_{p,total}$ exhibits a minimum. Additionally, increasing the evaporator pressure results in a decrease in $c_{p,total}$.

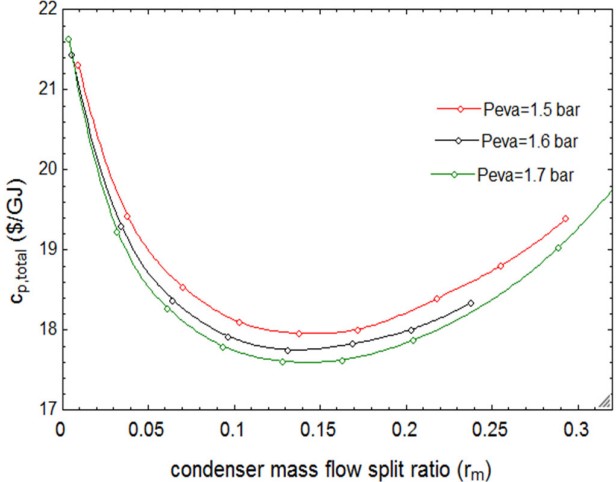

**Figure 14.** Effect of varying condenser mass flow split ratio on total product unit cost for several evaporator pressures.

Similarly, the effect of varying $r_m$ on $c_{p,total}$ is shown for several condenser pressures in Figure 15. It is evident that, as the condenser pressure rises, the value of $c_{p,total}$ increases considerably, contrary to Figure 14 where $c_{p,total}$ decreases with increasing evaporator pressure.

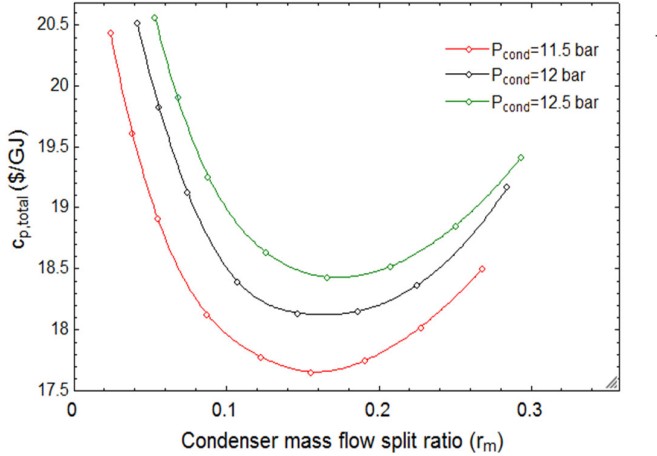

**Figure 15.** Effect of varying condenser mass flow split ratio on total product unit cost for several condenser pressures.

Figure 16 illustrates the effect of varying $r_m$ on $\eta_{exergy}$ for three condenser pressures. Note that there is a peak value for $\eta_{exergy}$ at each condenser pressure. Also, it is evident that $\eta_{exergy}$ is higher for a lower value of condenser pressure. The maximization of $\eta_{exergy}$ in Figure 16 can be explained by referring to Equation (12), considering that the input exergy rate is kept constant, while the $\dot{W}_{net}$ and $\dot{E}_{cooling}$ change as shown in Figure 17.

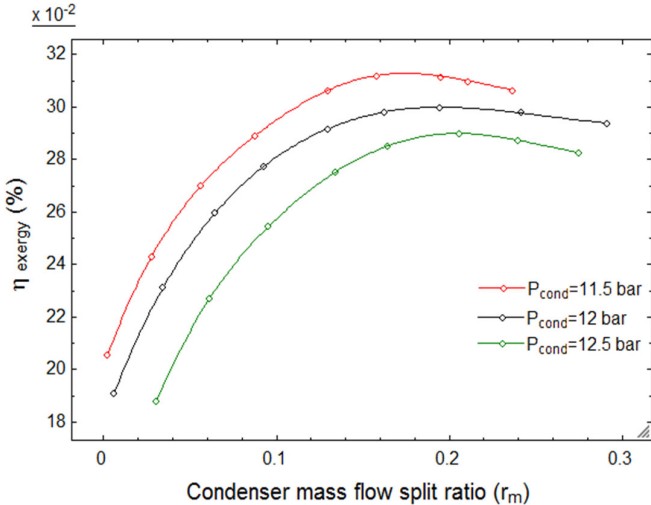

**Figure 16.** Effect of varying condenser mass flow split ratio on exergy efficiency for several condenser pressures.

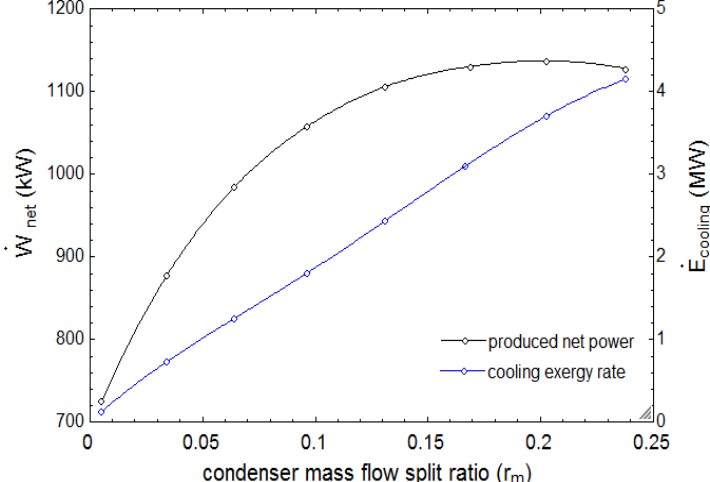

**Figure 17.** Effect of varying condenser mass flow split ratio on produced net power and cooling exergy rate.

The trend seen in Figure 17, with $\dot{W}_{net}$ exhibiting a peak, can be justified considering Figure 18, comparing the produced turbine power and its mass flow rate. As the $r_m$ increases, the amount of produced turbine power as well as the turbine mass flow rate follow opposite trends, explaining how $\dot{W}_{net}$ varies in Figure 17.

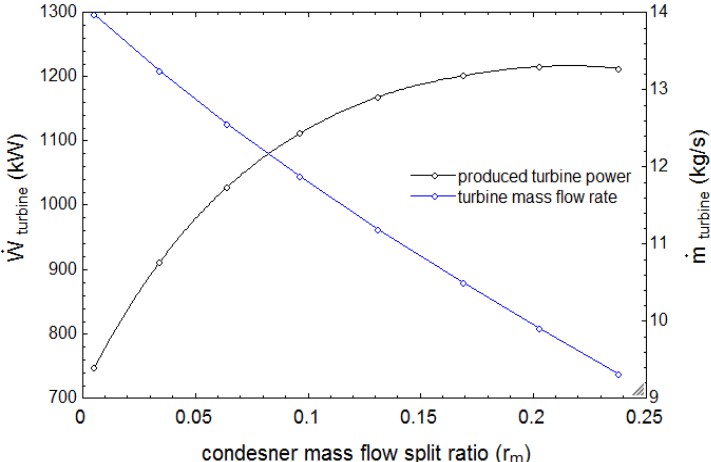

**Figure 18.** Effect of varying condenser mass flow split ratio on produced turbine power and turbine mass flow rate.

In the parametric study section, the effects of three important parameters are investigated on the system performance. It can be concluded Figures 3–18 that the $c_{p,total}$ reaches a minimum amount while these three parameters are varied separately. The lowest value for $c_{p,total}$, 17.3 $\$/\text{GJ}$, is achieved with an optimum value of evaporator pressure which is 5.5 bar. Increasing the amounts of evaporator and turbine inlet pressures results in a decrease of $c_{p,total}$, whereas the increase of condenser pressure results in a higher $c_{p,total}$ value.

For a condenser mass flow split ratio of $r_m = 0.15$ the exergy efficiency is maximized at a value of 31%. The exergy efficiency is also maximized $\eta_{exergy} = 29\%$ at a turbine inlet pressure of 31 bar.

## 7. Optimization

In the parametric study section, the effects of several parameters on the cycle performance are studied. However, only one parameter is considered in the study each time. In this section, all of the parameters are taken as variables to determine the optimum working condition for the system using the Variable Metric Optimization Method in the EES. This method is well known and commonly used in connection with unconstrained optimization, since it demonstrates good theoretical and practical convergence properties [33]. The basic idea is to fit the objective function to a quadratic function of all independent variables.

Two objectives are considered: the exergy efficiency and the total product unit cost. Hence, two separate cases are examined. In the first case, known as the exergy efficiency optimal design (EOD), the exergy efficiency is maximized and in the second case, known as the cost-optimal design (COD), the total product unit cost is minimized. The range of variations for the parameters influencing system performance are as follows:

$$18 \text{ bar} \leq P_{turbine} \leq 30 \text{ bar}$$

$$1.5 \text{ bar} \leq P_{evaporator} \leq 4 \text{ bar}$$

$$0.05 \leq r_m \leq 0.35$$

$$8 \text{ bar} \leq P_{condesner} \leq 13 \text{ bar}$$

$$5\,^{\circ}\mathrm{C} \leq \Delta T_{pp} \leq 15\,^{\circ}\mathrm{C}$$

$$100\,^{\circ}\mathrm{C} \leq T_{generator} \leq 135\,^{\circ}\mathrm{C}$$

The optimization results (local optimums) are summarized in Table 7, which compares performance parameters for the original, EOD, and COD cases.

**Table 7.** Optimization results.

| Variable | | Original Case | Optimization | |
|---|---|---|---|---|
| | | | **EOD** | **COD** |
| Condenser mass flow split ratio | $r_m$ | 0.07 | 0.21 | 0.15 |
| Turbine pressure ratio (TPR) | $TPR = P_{5a}/P_6$ | 2 | 2.8 | 2.5 |
| Generator temperature ($^{\circ}$C) | $T_{21}$ | 116 | 130 | 135 |
| Condenser pressure (bar) | $P_{condesner}$ | 12 | 12.6 | 11.8 |
| Evaporator pressure (bar) | $P_{evaporator}$ | 1.6 | 3.5 | 4 |
| Pinch Point temperature (°C) | $\Delta T_{pp}$ | 8 | 5 | 7 |
| Heat exchanger temperature drop (°C) | $\Delta T_{HE}$ | 10 | 11 | 9 |
| Net produced power ($\mathrm{kW}$) | $\dot{W}_{net}$ | 992 | 1154 | 1120 |
| Cooling capacity rate ($\mathrm{kW}$) | $\dot{Q}_{cooling}$ | 68 | 230 | 114 |
| Thermal efficiency (%) | $EUF$ | 6.8 | 14.7 | 9.2 |
| Exergy efficiency (%) | $\eta_{exergy}$ | 26.4 | 34.7 | 29.8 |
| Produced power unit cost ($\$/\mathrm{GJ}$) | $c_{p,power}$ | 13.7 | 11.8 | 11.1 |
| Cooling unit cost ($\$/\mathrm{GJ}$) | $c_{p,cooling}$ | 4.7 | 4.0 | 3.9 |
| Total product unit cost ($\$/\mathrm{GJ}$) | $c_{p,total}$ | 18.4 | 15.8 | 15.0 |

Referring to Table 7, the produced net power in the EOD case is 1154 kW which is 16.33% and 3.1% higher than the values obtained for the original and COD cases, i.e., 992 kW and 1120 kW, respectively. Likewise, the cooling capacity in the EOD case is 230 kW being 238.2% and 101.7% higher than the corresponding values obtained for the original and EOD cases, i.e., 68 and 114 kW, respectively. It is also apparent from Table 7 that the EOD case has the highest exergy efficiency (34.7%), which is 31% and 16% higher than the corresponding value for the original and COD cases, respectively. With respect to the thermal efficiency, again the EOD case exhibits about a 60% higher energy efficiency than the COD case. Regarding $c_{p,total}$, the COD case exhibits the lowest value among all cases; that value is 18.5% and 5.11% lower than the corresponding value for the original and EOD cases respectively. Also, it is apparent that, in the COD case, both power and cooling unit costs are lower than for the other cases.

Lastly, it can be seen that the turbine pressure ratio as well as condenser mass flow split ratio take on higher values for the EOD case compared to other cases.

Table 8 compares the amount of cooling and power unit costs for the cogeneration system, ARC, and KC under the original case, and the EOD as well as the COD cases. It is obvious that combining the KC and ARC results in reduced values for power and cooling unit costs in all cases.

Comparing the results obtained for individual cycles and the proposed cycle, it is observed that the power unit cost obtained with the cogeneration system is lower by of 12.4%, 11.9%, and 13.2% under original, EOD, and COD cases, respectively. The corresponding values for cooling unit cost are 26.5%, 23.0%, and 20.4%, respectively.

**Table 8.** Comparison of power and cooling unit costs.

| Cycle | Original Case | EOD | COD |
|:---:|:---:|:---:|:---:|
| **Cogeneration** | $c_{power} = 13.4 \; \$/GJ$ | $c_{power} = 11.8 \; \$/GJ$ | $c_{power} = 11.1 \; \$/GJ$ |
| | $c_{cooling} = 4.7 \; \$/GJ$ | $c_{cooling} = 4.0 \; \$/GJ$ | $c_{cooling} = 3.9 \; \$/GJ$ |
| KC | $c_{power} = 15.3 \; \$/GJ$ | $c_{power} = 13.4 \; \$/GJ$ | $c_{power} = 12.8 \; \$/GJ$ |
| ARC | $c_{cooling} = 6.4 \; \$/GJ$ | $c_{cooling} = 5.2 \; \$/GJ$ | $c_{cooling} = 4.9 \; \$/GJ$ |

## 8. Exergoeconomic Factor

In this section, the original, COD and EOD cases are investigated and compared from the viewpoint of thermoeconomics. This analysis identifies the most important components, i.e., those which have the highest exergy destruction rate and highest exergoeconomic factor ($f_k$). In Table 9, various parameters including exergy destruction rate ($\dot{E}_{D,k}$), cost rate of exergy destruction ($\dot{C}_{D,k}$), and exergoeconomic factor are calculated for components individually and compared for the different cases. It is observed that $\dot{E}_{D,total}$ has the highest value in the original case and is 10.2% and 10.5% higher than the corresponding values for the EOD and COD cases respectively. Also, it is notable that the COD case exhibits the lowest value of $\dot{C}_{D,total}$, whereas the original case exhibits the highest. For all cases, the Kalina boiler and two absorbers contribute the highest exergy destruction rates respectively, while the evaporator, mixers and pumps contribute the least. Additionally, the two absorbers and the boiler exhibit the highest values of $\dot{C}_{D,k}$.

In Table 9, all components are arranged in descending order based on the value of the sum $\dot{C}_{D,k} + \dot{Z}_k$. The results indicate that the Kalina turbine, absorber 2, and absorber 1 take on the first three places, which implies that more consideration in terms of optimization and modification should be paid to those components to improve their cost efficiencies. Regarding values for $f_k$ in Table 9 and Equation (22), the large values of the exergoeconomic factor are observed for the turbine, pumps, and RHE 2. This indicates that the capital investment and operating and maintenance costs dominate and they should be reduced in order to improve the cost effectiveness of the component. On the other hand, the relatively lower values of the factor for the absorbers and the boiler suggests a decrease in the exergy destruction value for those components would help to achieve to a more cost effective system.

**Table 9.** Exergeoeconomic analysis results for the original, EOD and COD cases.

| Component | Exergy Destruction Rate $\dot{E}_{D,k}$ (MW) | | | Exergy Destruction Cost Rate $\dot{C}_{D,k}$ ($/yr.) | | | Investment Cost Rate $\dot{Z}_k$ ($/yr.) | | | $\dot{C}_{D,k}+\dot{Z}_k$ ($/yr.) | | | Exergoeconomic Factor $f$ (%) | | |
|---|---|---|---|---|---|---|---|---|---|---|---|---|---|---|---|
| | Original Case | EOD | COD | Original Case | EOD | COD | Original Case | EOD | COD | Original Case | EOD | COD | Original Case | EOD | COD |
| Turbine | 0.167 | 0.194 | 0.189 | 2.84 | 2.68 | 2.65 | 30.7 | 32.5 | 32.1 | 33.5 | 35.2 | 34.7 | 91.5 | 92.4 | 92.3 |
| Absorber 2 | 0.298 | 0.343 | 0.265 | 10.1 | 9.32 | 7.47 | 16.7 | 9.76 | 14.1 | 26.8 | 19.1 | 21.6 | 62.2 | 51 | 65.4 |
| Absorber 1 | 0.342 | 0.371 | 0.36 | 10.1 | 10.3 | 10.3 | 12.89 | 4.64 | 3.6 | 22.9 | 15 | 13.9 | 56 | 30.8 | 25.8 |
| Boiler | 1.09 | 0.399 | 0.78 | 5.42 | 1.97 | 3.9 | 3.02 | 3.65 | 2.74 | 8.45 | 5.63 | 6.64 | 35.7 | 64.9 | 41.3 |
| AHE | 0.219 | 0.286 | 0.233 | 3.73 | 3.94 | 3.14 | 2.22 | 2.2 | 2.13 | 5.83 | 6.14 | 5.27 | 35.9 | 35.8 | 40.4 |
| SHE | 0.082 | 0.163 | 0.111 | 1.44 | 2.29 | 1.61 | 3.391 | 2.89 | 2.38 | 4.83 | 5.18 | 3.97 | 70.1 | 55.7 | 59.9 |
| Condenser | 0.043 | 0.088 | 0.044 | 0.733 | 1.22 | 0.62 | 2.09 | 4.45 | 2.82 | 3.64 | 5.67 | 3.44 | 79.8 | 78.3 | 81.9 |
| Generator | 0.067 | 0.08 | 0.094 | 1.19 | 1.12 | 1.34 | 2.15 | 2.9 | 2.56 | 3.34 | 4.03 | 3.9 | 64.3 | 72 | 65.6 |
| RHE 2 | 0.033 | 0.031 | 0.025 | 0.574 | 0.43 | 0.36 | 2.73 | 2.65 | 2.4 | 3.31 | 3.08 | 2.76 | 82.6 | 86 | 85.6 |
| Evaporator | 0.005 | 0.041 | 0.0056 | 0.096 | 0.571 | 0.08 | 2.65 | 2.34 | 1.96 | 2.74 | 2.91 | 2.04 | 96.5 | 80.3 | 96.1 |
| RHE 1 | 0.024 | 0.156 | 0.038 | 0.405 | 2.15 | 0.53 | 2.09 | 3.35 | 3.41 | 2.62 | 5.51 | 3.94 | 84.4 | 60.8 | 86.4 |
| Pump 1 | 0.003 | 0.006 | 0.006 | 0.067 | 0.092 | 0.08 | 0.78 | 1.05 | 0.91 | 0.85 | 1.14 | 1.03 | 92.1 | 91.9 | 91.4 |
| Pump 2 | 0.002 | 0.003 | 0.002 | 0.04 | 0.041 | 0.28 | 0.43 | 0.49 | 0.42 | 0.82 | 1.09 | 0.94 | 95 | 95.8 | 96.9 |
| Mixer 1 | 0.0001 | 0.0003 | 0.0003 | 0.06 | 0.022 | 0.02 | 0 | 0 | 0 | 0.06 | 0.02 | 0.02 | 0 | 0 | 0 |
| Mixer 2 | 0.0003 | 0.0002 | 0.0002 | 0.04 | 0.091 | 0.06 | 0 | 0 | 0 | 0.04 | 0.05 | 0.06 | 0 | 0 | 0 |
| **Total** | **2.38** | **2.16** | **2.15** | **36.6** | **36.2** | **32.1** | **82.7** | **73** | **71.6** | **119.8** | **109.8** | **104.3** | **946** | **894** | **929** |

## 9. Conclusions

In the current study, the proposed cycle generating cooling and power simultaneously is studied from thermodynamic and economic viewpoints. Energy, exergy, and exergy cost rates are calculated for all cycle streams. Under the base case conditions, the thermal and exergy efficiencies are achieved as 7.10% and 26.4%, respectively and the total product unit cost is calculated as 18.4 $/GJ.

The effects are investigated of three significant parameters, turbine inlet pressure (TIP), evaporator pressure and condenser mass flow split ratio, on the exergy efficiency and total product unit cost of the system. It is observed that as these three parameters are varied separately, $c_{p,total}$ achieves a minimum. It is concluded that an increase in the evaporator pressure and also in the turbine inlet pressure results in a decrease of $c_{p,total}$, while an increase in the condenser pressure raises the $c_{p,total}$ and decreases the exergy efficiency.

Comparing the influences of turbine inlet pressure, condenser pressure, and condenser mass flow split ratio the lowest $c_{p,total}$ value is achieved with the optimum value of evaporator pressure which is about 17.3 $/GJ for an evaporator pressure of 5.5 bar and a turbine inlet pressure of 24 bar. Furthermore, the exergy efficiency reaches its peak value (31% and 29%) at the optimum values of 0.15 and 31 bar for $r_m$ and TIP, respectively.

Following the parametric study, all of the previously mentioned variables plus others like generator temperature, condenser pressure and pinch point temperature difference are considered as variables to optimize the system to achieve the highest exergy efficiency (EOD case) or the lowest total product unit cost (COD case). The results show that the EOD case exhibits 60% and 16.4% higher energy and exergy efficiencies respectively than the COD case. However, the total product unit cost is 5.11% lower for the COD case than the EOD, at the expense of a 16.4% reduction in the exergy efficiency. Additionally, the EOD case produces a higher value of net power and cooling capacity than the COD case.

By comparing the proposed cogeneration system with its subsystems, it is seen that combining the two subsystems leads in reductions of 12.4%, 11.9%, and 13.2% in the power unit cost in the original, EOD, and COD cases, respectively compared to when power is produced from the Kalina Cycle individually. Furthermore, the unit cost of cooling has reduced by 26.5%, 23.0%, and 20.4% in the mentioned case studies in comparison with when ARC generates cooling separately.

With respect to the thermoeconomic analysis, the original case has the highest total values of exergy destruction rate as well as total exergy destruction cost rate. It can be seen that the highest exergy destruction and exergy destruction cost rates among the components is attributable to the boiler and absorbers respectively, whereas the evaporator, mixers, and pumps contribute to the lowest exergy destruction rates Also, the sum $\dot{C}_{D,k} + \dot{Z}_k$ is higher for the turbine and two absorbers, which implies that they merit more attention for reduction and optimized than other components, because these components have the highest rates of exergy destruction plus investment costs.

A relatively higher values of $f_k$ for turbine, pumps, and RHE2 suggests that a reduction in the cost of operating and maintenance and capital investment costs could be of benefit for the system to be more cost-efficient. On the other hand, reducing exergy destruction rates in components with a lower $f_k$ such as absorbers and boiler would improve the cost-efficiency of the system.

**Author Contributions:** All authors have read and agreed to the published version of the manuscript. The modeling was made by Javanshir; and analysis as well as discussion were performed by all the authors. All the authors contributed equally in writing the article. All authors have read and agreed to the published version of the manuscript.

**Funding:** This research received no external funding.

**Conflicts of Interest:** The authors declare no conflict of interest.

## Nomenclature

| | | | | |
|---|---|---|---|---|
| $\dot{C}$ | Cost rate ($/hr.) | | Eva | evaporator |
| $c$ | Unit exergy cost ($/GJ) | | Gen | generator |
| CI | Capital investment cost ($) | | OM | operating & maintenance |
| CRF | Capital recovery factor | | RHE | reheating heat exchanger |
| $\dot{E}$ | Exergy rate (kW) | | SHE | solution heat exchanger |
| $\dot{E}_D$ | Exergy destruction rate (kW) | | sep | separator |
| $e$ | Specific exergy (kJ/kg) | | Sp | spread point |
| $h$ | Specific enthalpy (kJ/kg) | | | |
| $i_r$ | Interest rate | | **subscripts** | |
| M | Molar mass | | 0 | Ambient condition |
| $\dot{m}$ | Mass flow rate (kg/s) | | $ch$ | Chemical |
| n | Number of operating years | | $e$ | output |
| P | Pressure (bar) | | $i$ | input |
| $r_m$ | Condenser mass flow split ratio | | $k$ | component |
| s | Specific entropy (kJ/kg.K) | | $ph$ | Physical |
| T | Temperature (K) | | | |
| U | Heat transfer coefficient $(KW/m^2.K)$ | | **Greek symbols** | |
| $\dot{W}$ | Work rate (power) (kW) | | | |
| $\dot{W}_{net}$ | Net work rate (power) (kW) | | $\gamma_k$ | Fixed operation cost coefficient |
| X | Ammonia mass concentration | | $\Delta T_{HE}$ | Heat exchanger temperature drop (°C) |
| Z | Investment cost ($) | | $\eta_{turbine}$ | Turbine isentropic efficiency |
| $\dot{Z}$ | Investment cost rate ($/hr.) | | $\eta_{pump}$ | Pump isentropic efficiency |

**Abbreviations**

| | | | | |
|---|---|---|---|---|
| | | | $\eta_{ex}$ | Exergy efficiency |
| Abs | absorber | | $R_k$ | Other operation costs ($) |
| ARC | absorption refrigeration cycle | | $\tau$ | Annual operation hours |
| AHE | absorption heat exchanger | | $\omega_k$ | Variable operation cost ($/MJ) |
| cond | condenser | | | |

## Appendix A. Calculating Equipment Cost Rate ($\dot{Z}_k$)

In order to assess the purchased equipment cost for the turbine, heat exchangers and pumps that are utilized in the cycle, the following equations are used in this study [34]:

$$Z_{turbine} = C_{P,turbine}\left(F_{M,turbine} \times F_{P,turbine}\right) \tag{A1}$$

$$Z_{pump} = C_{P,pump}\left(B_{1,pump} + B_{2,pump} \times F_{M,pump} \times F_{P,pump}\right) \tag{A2}$$

$$Z_{heat\ exchanger} = C_{P,heat\ exchanger}\left(B_{1,HE} + B_{2,HE} \times F_{M,HE} \times F_{P,HE}\right) \tag{A3}$$

where $C_p$ denotes the bare module cost for each component which can be calculated as follows [35]:

$$\log C_{p,x} = K_{1,x} + K_{2,x} \log Y + K_{3,x} (\log Y^2) \tag{A4}$$

The value $Y$ in Equation (A4) implies the area of the heat exchanger or the capacity of the turbine and pump. Also $K_1$, $K_2$, and $K_3$ are the coefficients of equipment cost. Their values can be found in Table A1. The material factor ($F_M$) and constants $B_1$, $B_2$, and $B_3$ are listed in Table A1 too. The relation for the pressure factors in the above equations ($F_P$) can be determined as follows [34]:

$$\log F_{p,x} = C_{1,x} + C_{2,x} \log(10P-1) + C_{3,x} (\log(10P-1))^2 \tag{25}$$

where pressure factors ($C_1$, $C_2$, and $C_3$) can be found in Table A1. Costs obtained from Equations (A1)–(A3) are in the reference year and need to be converted to the present year (2019). This is done by applying the Marshall and Swift equipment cost equation [34]:

$$Z_{\text{reference year}} = Z_{\text{original cost}} \times \frac{CI_{M.S}^{2019}}{CI_{M.S}^{\text{reference year}}} \tag{26}$$

**Table A1.** Equipment cost parameters.

| $X$ | $Y$ | $K_1$ | $K_2$ | $K_3$ | $B_1$ | $B_2$ | $C_1$ | $C_2$ | $C_3$ | $F_M$ |
|---|---|---|---|---|---|---|---|---|---|---|
| Turbine | $\dot{W}_{turbine}$ (kW) | 2.71 | 1.44 | −0.118 | 0 | 1 | 0 | 0 | 0 | 3.40 |
| Pump | $\dot{W}_{pump}$ (kW) | 3.39 | 0.0540 | 0.154 | 1.90 | 1.35 | −0.390 | 0.40 | −0.002 | 1.60 |
| Heat exchanger | $A_{heat\ exchanger}$ (m$^2$) | 4.33 | −0.300 | 0.164 | 1.63 | 1.66 | 0.0390 | −0.11 | 0.081 | 1.40 |

In the current study, all heat exchangers including the evaporator, condenser, absorbers, SHE, and AHE are regarded as shell-tube types and their areas can be calculated by the LMTD method and applying the following equation [35]:

$$\dot{Q}_k = A_k U_k \Delta T_k^{lm} \tag{27}$$

where $U_k$ denotes the overall heat transfer coefficient for each heat exchanger, values for which can be found in Table A2.

**Table A2.** Values of total heat transfer coefficient ($U_k$) for several heat exchangers.

| Component | $U_k$ ($\frac{W}{m^2 K}$) |
|---|---|
| Generator | 1.6 |
| Absorber | 1.1 |
| Evaporator | 0.9 |
| Condenser | 1.1 |
| Boiler | 0.9 |
| Heat exchangers (AHE,SHE,RHE) | 1 |

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
