# Peer review of "Energy and Cost Analysis and Optimization of a Geothermal-Based Cogeneration Cycle Using an Ammonia-Water Solution: Thermodynamic and Thermoeconomic Viewpoints"

_sustainability, doi:10.3390/su12020484_

Round 1
Reviewer 1 Report
System description can be improved, maybe it is better to use bullets to describe it. Figure 2 should be explained more. It represents validation for operation of the two systems, so, please, detail a bit more. Table 3 needs more explanations. Table 4 needs to be explained, e.g. yearly operation hours (8000 h/year) that means that system operates all year round and supplies both power and cooling, so, please, indicate potential consumer, since I assume that cooling is for air conditioning (specify it as well). Interest rate is bit high, especially when working with USD, so please explain it too. Tables 5 & 6 (especially 6) should be explained a bit more. For the chapter 5. Parametric study it might be useful to a paragraph at the end of the chapter summarizing all the findings. This chapter can be also included in the Conclusions. Table 7 should be explained more, for example in terms of power and cooling production for cases EOD & COD compared to original case, and also other parameters. Table 8 should be explained a bit more. Table 9 can be divided into several tables presenting for example exergy destruction rate for each component starting with the highest value so there can be seen which equipment contributes most to exergy destruction. The same can be done exergy destruction cost rate and for other parameters. Conclusion chapter should be improved by adding more information/results obtained in Parametric study, optimization and thermoeconomic analysis. Improve English.Author Response
Reviewer 1:
Dear reviewer 1,
We do appreciate your useful comments, which have helped us improve our manuscript. We have considered all the comments carefully and addressed them in the manuscript as appropriate. The comments and our responses are as follows:
System description can be improved, maybe it is better to use bullets to describe it.
Reply: The structure of this section is modified and some explanations are added to it (please see the system description section).
Figure 2 should be explained more. It represents validation for operation of the two systems, so, please, detail a bit more.
Reply: Some more explanations are added to the text, as requested (please see lines 214 to 217).
Table 3 needs more explanations.
Reply: Some more explanations are added to the text, as requested (please see page 9 lines 218 to 219).
Table 4 needs to be explained, e.g. yearly operation hours (8000 h/year) that means that system operates all year round and supplies both power and cooling, so, please, indicate potential consumer, since I assume that cooling is for air conditioning (specify it as well).
Reply: The produced power from the system is desired almost all around the year, so it justifies the amount of yearly operation hours. The cooling, however can be used for some industrial cooling process such as gas turbine inlet air cooling or as some part of refrigeration in food industry. A statement is added to the text in this regard (please see line 227-228)
Interest rate is bit high, especially when working with USD, so please explain it too.
Reply: The interest rate used in this study is adopted from several references in literature (see for example reference 24 and the following reference). However, the change in interest rate will not change the results qualitatively.
A new flexible geothermal based cogeneration system producing power and refrigeration, part two: The influence of ambient temperature, Renewable Energy, https://doi.org/10.1016/j.renene.2018.11.082Tables 5 & 6 (especially 6) should be explained a bit more.
Reply: Some more explanations are added to the text, as requested (please see lines 233 to 235 and 239 to 242).
For the chapter 5. Parametric study it might be useful to a paragraph at the end of the chapter summarizing all the findings. This chapter can be also included in the Conclusions.
Reply: A paragraph summarizing the most important findings in the parametric study section is added to the text, as requested (please see lines 351 to 363).
Table 7 should be explained more, for example in terms of power and cooling production for cases EOD & COD compared to original case, and also other parameters.
Reply: An explanation is added to the text in this regard, as requested (please see lines 386 to 390).
Table 8 should be explained a bit more.
Reply: An explanation is added to the text, as requested (please see lines 402 to 405).
Table 9 can be divided into several tables presenting for example exergy destruction rate for each component starting with the highest value so there can be seen which equipment contributes most to exergy destruction. The same can be done exergy destruction cost rate and for other parameters.
Reply: We have followed the procedure suggested by Bejan et al. as it is practiced in literature, see for example references 24.
Conclusion chapter should be improved by adding more information/results obtained in parametric study, optimization and thermoeconomic analysis. Improve English.
Reply: More results are added to the conclusion section from different parts, as requested, please see the conclusion section, lines 438 to 445. The manuscript is revised again and the English is improved.

Reviewer 2 Report
This study presents a novel modified Kalina cycle which produces power and absorption refrigeration cycle. In addition, this study also conducted optimization by using two different objective functions such as the exergy efficiency and the total unit cost.
The topic covered in this study may be interesting to readers, but some points mentioned below must be clearly answered before publication in this journal.
Introduction and System Description section Authors insist on the proposed Kalina cycle in this study is novel, I can not find the novelty of the proposed cycle. The authors should clearly indicate the novelty of the proposed Kalina cycle in Introduction and System Description. Parametric Study I think this section should be totally modified. The parametric study should be performed after optimization to recognize which parameters (or Boundary conditions) can affect the results of optimization. However, the parametric study performed in this study used the same variables used in the optimization section. This is definitely not necessary. I recommend you to perform some parametric studies regarding changes in environmental temperature, geothermal inlet temperature, geothermal rejection temperature, interest rate or what the authors consider important. Optimization What optimization algorithm was used for this study? And why did the authors use that kind of optimization algorithm? The result presented in this study is global optimum or local optimum?Author Response
Reviewer 2:
Dear reviewer 2,
We do appreciate your useful comments, which have helped us improve our manuscript. We have considered all the comments carefully and addressed them in the manuscript as appropriate. The comments and our responses are as follows:
Introduction and System Description section: Authors insist on the proposed Kalina cycle in this study is novel, I cannot find the novelty of the proposed cycle. The authors should clearly indicate the novelty of the proposed Kalina cycle in Introduction and System Description.
Reply: The Kalina cycle, in the present work, is a modified Kalina cycle and not a novel one, as outlined in the abstract (lines 15-16). The proposed cycle is a combination of the modified Kalina cycle and absorption refrigeration cycle in a novel way so that they have only one condenser. This has been explained in the system description section (lines 98 to 100).
Parametric Study: I think this section should be totally modified. The parametric study should be performed after optimization to recognize which parameters (or Boundary conditions) can affect the results of optimization. However, the parametric study performed in this study used the same variables used in the optimization section. This is definitely not necessary. I recommend you to perform some parametric studies regarding changes in environmental temperature, geothermal inlet temperature, geothermal rejection temperature, interest rate or what the authors consider important.
Reply: In the parametric study the influences of different parameters on the system performance are investigated to find out the main variables affecting the system performance so that the optimum value of these variables is determined by the optimization process. This procedure has been practiced in literature, see reference 24 of the present work and the following reference as examples:
A new flexible geothermal based cogeneration system producing power and refrigeration, part two: The influence of ambient temperature, Renewable Energy, https://doi.org/10.1016/j.renene.2018.11.082Regarding the variations in environmental temperature, geothermal inlet temperature, geothermal rejection temperature, and interest rate your suggestion is greatly appreciated and we will work on them in our future works.
Optimization: What optimization algorithm was used for this study? And why did the authors use that kind of optimization algorithm? The result presented in this study is global optimum or local optimum?
Reply: The Variable Metric Optimization Method, as included in the EES, is used in the present work for optimization. Explanations regarding this method and some references are added to the text in the optimization section (please see lines 365 to 370). The results of optimization, in the present work, are local optimums.

Round 2
Reviewer 2 Report
All my concerns are well addressed.